# Binary Promoter Improving the Moderate-Temperature Adhesion of Addition-Cured Liquid Silicone Rubber for Thermally Conductive Potting

**DOI:** 10.3390/ma15155211

**Published:** 2022-07-28

**Authors:** Jia-Kai Wu, Kai-Wen Zheng, Qiong-Yan Wang, Xin-Cheng Nie, Rui Wang, Jun-Ting Xu

**Affiliations:** 1MOE Key Laboratory of Macromolecular Synthesis and Functionalization, Department of Polymer Science & Engineering, Zhejiang University, Hangzhou 310027, China; 11429008@zju.edu.cn (J.-K.W.); 15117965820@163.com (K.-W.Z.); xujt@zju.edu.cn (J.-T.X.); 2Research and Development Center, Zhejiang Sucon Silicone Co., Ltd., Shaoxing 312088, China; sxyxnxc@163.com (X.-C.N.); wangrhyc88@163.com (R.W.)

**Keywords:** binary promoter, liquid silicone rubber, thermally conductive, moderate-temperature adhesion

## Abstract

The strong adhesion of thermally conductive silicone encapsulants on highly integrated electronic devices can avoid external damages and lead to an improved long-term reliability, which is critical for their commercial application. However, due to their low surface energy and chemical reactivity, the self-adhesive ability of silicone encapsulants to substrates need to be explored further. Here, we developed epoxy and alkoxy groups-bifunctionalized tetramethylcyclotetrasiloxane (D_4_H-MSEP) and boron-modified polydimethylsiloxane (PDMS-B), which were synthesized and utilized as synergistic adhesion promoters to provide two-component addition-cured liquid silicone rubber (LSR) with a good self-adhesion ability for applications in electronic packaging at moderate temperatures. The chemical structures of D_4_H-MSEP and PDMS-B were characterized by Fourier transform infrared spectroscopy. The mass percentage of PDMS-B to D_4_H-MSEP, the adhesion promoters content and the curing temperature on the adhesion strength of LSR towards substrates were systematically investigated. In detail, the LSR with 2.0 wt% D_4_H-MSEP and 0.6 wt% PDMS-B exhibited a lap-shear strength of 1.12 MPa towards Al plates when curing at 80 °C, and the cohesive failure was also observed. The LSR presented a thermal conductivity of 1.59 W m^−1^ K^−1^ and good fluidity, which provided a sufficient heat dissipation ability and fluidity for potting applications with 85.7 wt% loading of spherical α-Al_2_O_3_. Importantly, 85 °C and 85% relative humidity durability testing demonstrated LSR with a good encapsulation capacity in long-term processes. This strategy endows LSR with a good self-adhesive ability at moderate temperatures, making it a promising material requiring long-term reliability in the encapsulation of temperature-sensitive electronic devices.

## 1. Introduction

Optical, electronic devices and electrical modules are continuously evolving in functionality and achieving high performance while tending to be smaller and more integrated [1,2,3,4]. These devices are generally encapsulated with polymeric potting materials to achieve heat transfer and avoid environmental damages (e.g., moisture penetration and mechanical damage), thus boosting their long-term reliability [5,6,7]. Addition-cured liquid silicone rubber (LSR) possesses an amazing array of properties including a high dielectric breakdown strength, flame resistance, heat and cold resistance, stress relieving properties and long-lasting durability, which make it one of the most promising materials for the protection of electronic devices [8,9]. The poor mechanical properties of pure LSR limit its practical applications [9]. Researchers have expended great efforts to incorporate reinforcement fillers and functional fillers into LSR to improve its mechanical properties and introduce additional properties such as thermal conductivity and electrical conductivity [10,11,12,13]. LSR can be imparted with suitable thermal conductivity for efficient heat dissipation by doping with high thermal conductivity fillers such as carbon-based fillers, boron nitride, aluminum nitride, aluminum oxide (Al_2_O_3_) and so on [12,13,14,15,16].

Hydrophobic spherical α-Al_2_O_3_-contained LSR (LSR/Al_2_O_3_) achieves a high thermal conductivity (>1.0 W m^−1^ K^−1^) and high fluidity while maintaining a low viscosity (<10,000 mPa∙S), which are key properties for workability during the encapsulation [16,17,18]. Thus, LSR/Al_2_O_3_ composites have been widely used in the application of electronic devices encapsulation. However, addition-cured LSR presents a low surface energy and chemical reactivity, leading to a weak adhesion ability towards substrates [19,20]. Such weakly bonded interface is vulnerable to moisture erosion or damage from external forces, which can deteriorate the encapsulation effect of LSR in electronic devices operating in harsh environments [21,22,23,24]. Considerable efforts have been made to enhance the adhesion ability of LSR towards substrates, including primer pretreatment, substrate surface modification, the addition of an adhesion promoter, etc. [25,26,27,28,29]. Primers consisting of highly reactive molecules and solvents exhibit film-forming properties, good chemical reactivity and excellent wettability to most substrates, which are beneficial to enhancing the interfacial adhesion ability of LSR [25]. Grard et al. utilized xylene-diluted silane coupling agent mixtures as a primer to treat an AA6061 aluminum alloy surface to enhance its adhesion strength to silicone rubber [26]. The silicone rubber/aluminum alloy assembly showed a peel strength of 6.7 N m^−1^ and a 100% cohesive failure during a 90°-peel test. The substrate surface modification is an effective method to improve the adhesion strength of LSR to other materials by generating new reactive groups on the substrate surface. Roth et al. performed oxygen and ammonia plasma treatments on silicone rubber and the subsequent grafting of poly(ethylene-alt-maleic anhydride) to form a functionalized surface [27]. The modified silicone rubber showed improved permanent adhesion with epoxy resin and yielded a pull-off strength of 4.7 MPa. The above-mentioned methods can effectively improve the adhesion of LSR to various substrates; however, both methods are time-consuming and complicated processes, and they are harmful to the environment. Therefore, the addition of adhesion promoters into LSR to provide a self-adhesive capacity has spurred intensive research interest due to the simple and scalable process and eco-friendly properties [29,30,31,32,33]. Pan et al. reported a vinyl and epoxy groups-functionalized silane oligomer and used it as an adhesion promoter to prepare a self-adhesive addition-cure silicone encapsulant [32]. The LSR/Al_2_O_3_ encapsulant showed a shear strength of 1.06 MPa towards the Al substrate with a 2 wt% addition of an adhesion promoter after curing at 130 °C, which was about 2.12 times higher than that of LSR without adding an adhesion promoter. Epoxy, alkoxy and acrylate groups-modified prepolymer was synthesized by Wang et al. and was utilized as an adhesion promoter to improve the adhesion strength between LSR and the Al substrate [34]. The adhesion strength was effectively improved by 161% by incorporating 10.0 wt% of the prepared adhesion promoter in LSR. These reactive groups (i.e., epoxy, alkoxy and methacryloxy groups)-containing adhesion promoters generally demonstrated their great improvement in bonding performance at high working temperatures (>120 °C). However, most of the adhesion promoters are unable to offer an efficient adhesion ability for LSR at moderate curing temperatures (e.g., 80 °C) [31,32,33]. Therefore, it remains a challenge to achieve efficient encapsulation and provide long-term reliable protection for temperature-sensitive electronic devices.

The inefficiency of adhesion promoters is mainly derived from the low reactivity of epoxy groups at moderate temperatures [35]. In this work, an additional co-adhesive is designed to increase the reactivity of epoxy groups and to impart a good self-adhesion ability for applications in electronic packaging at moderate temperature. To this end, allyl dioxaborane-modified polydimethylsiloxane (PDMS-B) was synthesized and served as a co-adhesive to collaborate with olefin-based epoxy and alkoxy groups-bifunctionalized tetramethylcyclotetrasiloxane (D_4_H-MSEP) so as to offer the two-component addition-cured LSR with a good self-adhesion ability at moderate temperatures. The olefin-containing reactive groups and siloxane backbones endowed the adhesion promoters with a good compatibility with LSR, while the boron-containing groups were designed to facilitate the curing of epoxy groups. By incorporating two components, PDMS-B and D_4_H-MSEP, into LSR together, the as-prepared two-component addition-curing LSR presented good stability during storage and exhibited high reactivity and adhesion. The effect of the weight percentage of PDMS-B to D_4_H-MSEP and the content of adhesion promoters on the adhesion strength of LSR were systematically investigated. Additionally, the durability test at a temperature of 85 °C and a relative humidity of 85% was performed to evaluate the long-term stability of LSR and its encapsulation capacity.

## 2. Materials and Methods

### 2.1. Materials

Allyltrimethoxysilane (ATMS, >97.0%), 1,2-epoxy-5-hexene (EPHE, >96.0%), 2-allyl-4,4,5,5-tetramethyl-1,3,2-dioxaborolane (ATDB, ≥95%), 2,4,6,8-tetramethylcyclotetrasiloxane (D_4_H, 98%) and 3-methyl-1-pentyn-3-ol (98%), toluene (GR) were all purchased from Macklin Inc. Shanghai, China (www.macklin.cn, accessed on 1 February 2022). Vinyl-terminated polydimethylsiloxane with a reduced volatility (PDMS-Vi, viscosity of ~230 mPa∙s, product code: HYC-Vi230E) and hydride-terminated poly(methylhydro-co-dimethylsiloxane) (PHMS-H, viscosity of ~10 mPa∙s, product Code: HYC-LH-0.5) were obtained from Zhejiang Sucon Silicone Co., Ltd., Shaoxing, China (www.hycsilicon.com, accessed on 1 June 2022). Monovinyl-terminated polydimethylsiloxane (PDMS-mVi, asymmetric, 500 mPa∙s, product code: MCR-V25) was obtained from Gelest, Inc., Morrisville, PA, USA (www.gelest.com, accessed on 1 November 2021). Spherical α-Al_2_O_3_ modified by a silane coupling agent (BAH series) was acquired from Shanghai Bestry Performance Materials Co., Ltd., Shanghai, China (www.bestry-tech.com, accessed on 1 November 2021). The Karstedt’s platinum catalyst with a platinum content of 5000 ppm was obtained from Heraeus Materials Technology Shanghai Ltd., Shanghai, China (www.heraeus.cn, accessed on 1 June 2021).

### 2.2. Synthesis of Boron-Modified PDMS

Boron-modified PDMS (PDMS-B) was synthesized through a hydrosilylation reaction between ATDB and PHMS-H, as shown in Figure 1 Route A. The detailed synthesis procedure was as follows: PHMS-H (10.0 g) was dissolved in anhydrous toluene (20.0 mL) in a four-neck round bottom flask equipped with a temperature sensor, a dropping funnel, a reflux condenser and nitrogen-blow devices. The reaction solution was kept at 20 °C for 30 min to reach the pre-equilibration, followed by adding Karstedt’s platinum catalyst (0.02 g) into the solution. Subsequently, excessive ATDB (10.0 g) was added dropwise into the flask through the dropping funnel in 10 min, the reaction mixture was stirred for 2 h and then temperature was increased to 60 °C until all Si-H groups were reacted. The unreacted ATDB and toluene were removed through a thin-film evaporator at 60 °C and 200 Pa to yield the desired product PDMS-B as a colorless, transparent and viscous liquid.

### 2.3. Synthesis of Epoxy and Alkoxy Groups-Bifunctionalized D_4_H

The synthetic pathway of epoxy and alkoxy groups-bifunctionalized D_4_H is illustrated below (Figure 1 Route B): typically, ATMS (6.48 g, 40.0 mmol) and EPHE (3.92 g, 40.0 mmol) were dissolved in anhydrous toluene (20.0 mL) in the aforementioned four-neck round bottom flask. After adding Karstedt’s platinum catalyst (0.01 g), the flask was pre-equilibrated with N_2_ for 30 min at 20 °C. D_4_H (9.6 g, 40 mmol) was added dropwise into the solution for 10 min under vigorous stirring. After all of the –CH = CH_2_ groups from ATMS and EPHE were reacted, which was monitored by Fourier-transform infrared spectroscopy (stretching vibration of –CH = CH_2_ groups: 1640 cm^−1^), the solution was mixed with activated carbon for 2 h and filtrated through a 0.22 μm filter to remove the residual platinum catalyst. The unreacted monomers and toluene were removed through the thin-film evaporator to yield the epoxy and alkoxy bifunctional D_4_H-MSEP as a colorless and transparent liquid.

### 2.4. Preparation of Self-Adhesive Two-Component Addition-Curing LSR

PDMS-Vi, PDMS-mVi and Al_2_O_3_ were kneaded with a planetary mixer for 30 min in the designed amounts, as shown in Table 1, followed by a heat treatment at 150 °C and a vacuum of 500 Pa for 1 h to promote the uniform dispersion of Al_2_O_3_ in the PDMS matrix. Subsequently, the Karstedt’s platinum catalyst and PDMS-B were added in the cooled PDMS/Al_2_O_3_ matrix and kneaded for 30 min under vacuum conditions to obtain the component A of the two-component LSR. PDMS-Vi, PDMS-mVi and Al_2_O_3_ were treated as above to obtain uniform liquid silicone rubber blends. PHMS-H, D_4_H-MSEP and 3-methyl-1-pentyn-3-ol (platinum inhibitor) were mixed with a cooled PDMS/Al_2_O_3_ matrix under vacuum conditions to obtain the silicone composite termed as the component B of the two-component LSR. Component A and component B were thoroughly stirred and then mixed in equal proportions to afford the two-component LSR. The LSR was deaired at a vacuum of 500 Pa for 2 min before using.

### 2.5. Characterization

#### 2.5.1. Fourier Transform Infrared (FTIR) Analysis

FTIR measurements were performed on a Thermofisher Scientific Nicolet iS10 spectrometer (USA) to identify the chemical structures of the synthesized PDMS-B and D_4_H-MSEP. Samples (0.2 mL) were coated on KBr slices (25 mm × 2 mm) to form thin films for FTIR measurements.

#### 2.5.2. Viscosity Measurement

The viscosity of PDMS/Al_2_O_3_ composites was measured using a Brookfield DV2T viscometer at 25 °C according to the ASTM D-445 standard. Three specimens were measured for each measurement, and the mean value was calculated.

#### 2.5.3. Mechanical Properties Test

The tensile strength and elongation at break of the cured LSR samples were measured by a Gotech AI-7000S universal testing machine at a stretching rate of 50 mm min^−1^ at 25 °C according to the GB/T 528 standard. Under 3000 MPa pressure, LSR samples were cured at 80 °C for 2 h and then at 25 °C for another 24 h to form a 2 mm-thick rectangular sheet. The samples were tailored into dumbbell shapes before mechanical tests. Five specimens were measured for each measurement, and the mean value was calculated.

#### 2.5.4. Thermal Conductivities Test

The thermal conductivities of the LSR sheets (25 mm × 25 mm, 2 mm-thick) were tested by a thermal analyzer (TC3000E, Xiatech, Xi’an, China) according to the ASTM D-5930 standard. Five specimens were measured for each measurement, and the mean value was calculated

#### 2.5.5. Adhesion Performance

Tensile lap-shear strength tests were performed using a Gotech AI-7000S universal testing machine to evaluate the adhesion performance of the LSR towards Al and printed circuit board (PCB) plates (Figure 2), both of which are the major substrate materials for electronic devices. The Al and PCB plates were rinsed with isopropanol in an ultrasonic bath for 10 min at room temperature and then dried at 60 °C. Two-component LSR was potted into the tailored gap between two adherends and cured at 80 °C without any pressure followed by 25 °C for 24 h. The effective bonding surface of the LSR and adherends was 25 × 25 mm^2^, and the thickness of the LSR was 2 mm. The lap-shear strength between the LSR and adherends was calculated as the following equation:*τ**_s_* = *F**_m_*/*w*(1)
where *τ**_s_*, *F_m_* and *w* are the lap-shear strength, maximum force and effective bonding surface, respectively. Five specimens were measured for each measurement, and the mean value was calculated.

## 3. Results

### 3.1. Structural Characterization of PDMS-B and D_4_H-MSEP

The chemical structures of the raw materials and resultant PDMS-B and D_4_H-MSEP were identified by FT-IR, as shown in Figure 3. In Figure 3a, the characteristic band of 1300–1500 cm^−1^, with the peak centered at 1370 cm^−1^ and the sharp peak at 1640 cm^−1^, corresponded to the stretching vibration of B-O bonds and –CH = CH_2_ groups in ATDB, respectively [36,37,38]. The band in the range of 2080–2200 cm^−1^ was ascribed to the stretching vibration of -Si-H groups in PHMS-H. In the FT-IR spectrum of PDMS-B, the appearance of a new absorption peak for B-O bonds and the absence of an absorption peak for the -Si-H groups and –CH = CH_2_ groups indicated the complete hydrosilylation reaction between PHMS-H and ATDB and the successful synthesis of boron-modified PDMS. In Figure 3b, the sharp absorption peak at 2170 cm^−1^ was assigned to -Si-H groups in D_4_H. The characteristic absorption peaks at 2840 cm^−1^ and 910 cm^−1^ were ascribed to the stretching vibration of -Si-OCH_3_ groups in the ATMS and epoxy groups in EPHE, respectively [39,40,41]. The appearance of new absorption peaks for -Si-OCH_3_ groups and epoxy groups, the reduced absorption intensity of the peak for -Si-H groups and the disappearance of the absorption peak for –CH = CH_2_ groups in the yielded D_4_H-MSEP demonstrated that the epoxy groups and alkoxy groups-bifunctionalized D_4_H was synthesized. Notably, partial Si-H groups were intentionally retained, as observed in the spectrum of D_4_H-MSEP, in order to bind to the LSR matrix through hydrosilylation during the further curing process.

### 3.2. Thermal Conductivity and Mechanical Properties of LSR

The Al_2_O_3_ loading amount plays a key role in the thermal conductivity and viscosity of LSR, which directly affects the heat dissipation ability and potting processability. As shown in Figure 4, the thermal conductivity and viscosity of LSR monotonically increased with the increase in the Al_2_O_3_ content, and such an increasing trend is more obvious at a higher Al_2_O_3_ loading amount. In detail, the thermal conductivity of LSR increased from 0.78 W m^−1^ K^−1^ to 1.81 W m^−1^ K^−1^ as the Al_2_O_3_ content increased from 75 wt% to 87.5 wt%. The viscosity of LSR increased from 2622 mPa∙s to 9185 mPa∙s as the Al_2_O_3_ content increased from 75 wt% to 85.7 wt%. However, the viscosity increased dramatically to 14,452 mPa∙s when the Al_2_O_3_ content increased to 87.5 wt%. The aggregation of Al_2_O_3_ and the intermolecular interaction between the Al_2_O_3_ agglomerates resulted in the rapidly increased viscosity of LSR, which made it unsuitable for potting applications in some complex devices where the viscosity should be less than 10,000 mPa∙s. Therefore, the LSR with an 85.7 wt% Al_2_O_3_ loading amount, showing a thermal conductivity of 1.59 W m^−1^ K^−1^ and a viscosity of 9185 mPa∙s, was selected for the further improvement of the adhesion performance.

The effect of the Al_2_O_3_ content on the mechanical properties of the LSR were investigated and are shown in Figure 5. It was observed that the tensile strength and the elongation at break of the LSR were improved by the incorporation of the semi-reinforcing filler Al_2_O_3_. The optimal tensile strength of LSR reached up to 1.37 MPa with 83.3 wt% Al_2_O_3_ content, while the highest elongation at break reached up to 144.7% with 80.0 wt% Al_2_O_3_ content. However, the excessive incorporation of Al_2_O_3_ led to their aggregation, and the Al_2_O_3_ agglomerates act as defects in the LSR, reducing the effective interaction area between the particle and the matrix and hence the deterioration of the LSR mechanical properties. The LSR loaded with 85.7 wt% Al_2_O_3_ content, showing a tensile strength of 1.31 MPa and an elongation at break of 114%, was selected for the following studies.

### 3.3. Adhesion Performance of LSR

Figure 6 presented the effect of PDMS-B and D_4_H-MSEP content on the adhesion performance of LSR towards Al and PCB plates. The LSR containing 2.0 wt% D_4_H-MSEP showed a weak bond strength towards both Al and PCB substrates after curing at 80 °C, which could be almost removed from the substrates (Figure 6b). Upon the addition of PDMS-B, however, both the LSR/Al and LSR/PCB joints (D_4_H-MSEP content maintained at 2.0 wt%) showed remarkable enhancement in the lap-shear strength as the mass percentage of PDMS-B to D_4_H-MSEP increased from 0 to 40%. In detail, the LSR/Al joints showed a lap-shear strength of 1.16 MPa as the mass percentage of PDMS-B to D_4_H-MSEP approached 40%, which was 2.74 times higher than that of the LSR/Al joints (0.31 MPa) without adding PDMS-B. Moreover, the cohesive failure on the macroscale occurred at the interface of the LSR/Al joints when the mass percentage of PDMS-B to D_4_H-MSEP reached up to 30% (Figure 6b). Further increasing the mass percentage of PDMS-B only led to a slight improvement in the lap-shear strength of the LSR/Al joints. Thus, the 30% mass percentage of PDMS-B to D_4_H-MSEP can provide LSR with a sufficient adhesion ability. A similar enhancing trend of adhesion strength was also observed in LSR/PCB systems with increases in the PDMS-B content in LSR. These results demonstrated that the addition of PDMS-B indeed improved the adhesion ability of D_4_H-MSEP-modified LSR. The boron atoms in PDMS-B promoted the reaction between epoxy groups and hydroxyl groups on the substrates [42]. The hydrolysis of the dioxaborolane groups into boron hydroxyl groups during the curing process facilitated the strong interactions between the polar groups from the substrates and the alkoxy groups from the D_4_H-MSEP. By incorporating PDMS-B (in component A) and D_4_H-MSEP (in component B) into two components, the as-prepared addition-curing LSR presented good stability during storage and exhibited high reactivity and adhesion during the curing process.

The effect of D_4_H-MSEP content on the lap-shear strength of LSR and adherends joints was investigated and is shown in Figure 7. Without the addition of an adhesion promoter, the LSR showed almost no adhesion to adherends due to its low surface energy and low surface activity, leading to the easy removal of LSR from the adherend surfaces. In contrast, the lap-shear strength between the LSR and Al (PCB) plates significantly increased to 1.04 MPa as the D_4_H-MSEP content increased to 1.5 wt% (mass percentage of PDMS-B to D_4_H-MSEP was kept at 30%), followed by a smooth enhancement approaching 1.12 MPa with the further increase in D_4_H-MSEP content to 2.0 wt%. Increasing the adhesion promoter content would facilitate the enrichment of PDMS-B and D_4_H-MSEP molecules on the LSR surface, hence improving the adhesion strength of LSR. The concentration of the adhesion promoter molecules at the interface was limited owing to the steric hindrance; however, the lap-shear strength showed a limited enhancement when the adhesion promoters content exceeded 1.5 wt%. Overall, these results confirmed that the high reactivity and interface enrichment of the binary adhesion promoters PDMS-B and D_4_H-MSEP provided a synergistic effect on the enhancement of LSR adhesion performance and endowed the LSR with a great adhesion ability at a moderate temperature of 80 °C.

Figure 8 presents the effect of the curing temperature on the adhesion strength of LSR with adherends. The LSR exhibited a gradually enhanced lap-shear strength towards both Al and PCB plates as the curing temperature increased from 60 to 90 °C, resulting from the faster diffusion of the adhesion promoters to the LSR surface and the higher reactivity of the epoxy groups in D_4_H-MSEP at higher curing temperatures. When the curing temperature reached 80 °C, the lap-shear strength of the LSR towards both of the two adherends was higher than 1.0 MPa, and the cohesive failure was observed at the interface of the LSR/adherends joints after lap-shear strength testing. This result indicated that the obtained LSR could be used for the encapsulation of temperature-sensitive materials such as plastic seals and sensitive electronic components. The LSR encapsulant prepared in this work is compared against the addition-cured silicone encapsulants reported in the literature and commercial products (Table 2). The LSR potting compound described in this paper can achieve effective adhesion at lower temperatures.

The encapsulation with silicone potting materials can improve the device operation reliability and durability, which is critical for optical and electronic applications including fiber optic bundles, sensors, insulated gate bipolar transistors and so on [44]. The temperature of 85 °C and the relative humidity of 85% were set to carry out the durability testing so as to accelerate the evaluation of the device performance. The dependence of lap-shear strength on testing time was investigated to evaluate the long-term reliability of LSR for device encapsulation [45]. As shown in Figure 9, the lap-shear strength of LSR towards both Al and PCB plates was almost unchanged during the 200 h testing period. This good, durable adhesion would provide long-term protection for a variety of optical and electronic devices.

## 4. Conclusions

Binary adhesion promoters, boron-modified PDMS-B and epoxy and alkoxy groups-bifunctionalized D_4_H-MSEP, were synthesized in order to provide an addition-cured thermally conductive LSR encapsulant with a strong self-adhesion ability at moderate temperatures. The thermal conductivity, viscosity and mechanical and adhesion properties of the LSR encapsulant were systematically investigated. In detail, the LSR containing 2.0 wt% D_4_H-MSEP and 0.6 wt% PDMS-B showed a lap-shear strength of 1.12 MPa towards the Al plate when curing at 80 °C. The cohesive failure occurred at the interfaces of both the LSR/Al and LSR/PCB joints. The LSR with an 85.7 wt% Al_2_O_3_ loading content provided a sufficient heat dissipation ability and fluidity for potting applications. Importantly, durability testing at the temperature of 85 °C and at 85% relative humidity suggested that LSR features a good encapsulation capacity during a long-term operation. This design strategy endows LSR with long-term reliability for further potential applications in the encapsulation of temperature-sensitive optical and electronic devices at specific working temperatures.

## Figures and Tables

**Figure 1 materials-15-05211-f001:**
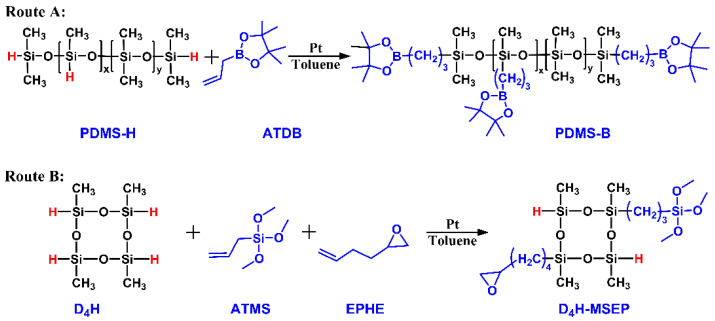
Schematic diagram for the synthesis of (**A**) PDMS-B and (**B**) D_4_H-MSEP.

**Figure 2 materials-15-05211-f002:**
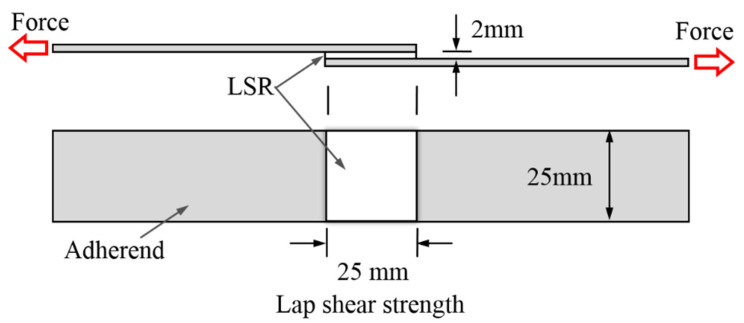
Schematic diagram of the tensile lap-shear strength test between the LSR and adherends.

**Figure 3 materials-15-05211-f003:**
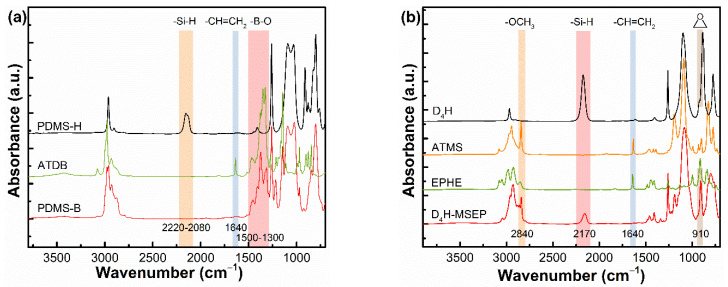
FT-IR spectra of (**a**) PDMS-H, ATDB and PDMS-B and (**b**) D_4_H, ATMS, EPHE and D_4_H-MSEP.

**Figure 4 materials-15-05211-f004:**
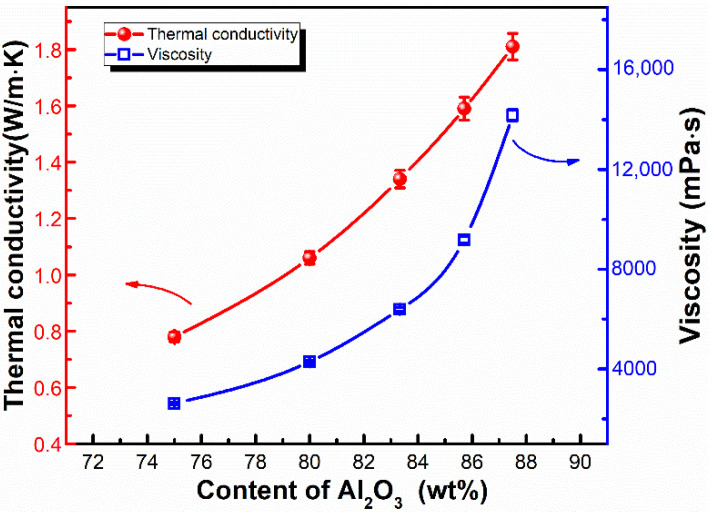
Dependences of the thermal conductivity and viscosity on the Al_2_O_3_ content in the LSR.

**Figure 5 materials-15-05211-f005:**
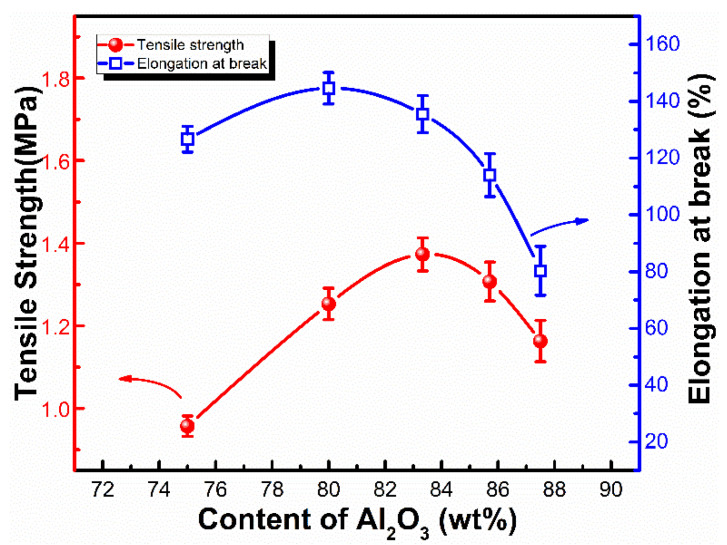
The mechanical properties of LSR with varied Al_2_O_3_ contents.

**Figure 6 materials-15-05211-f006:**
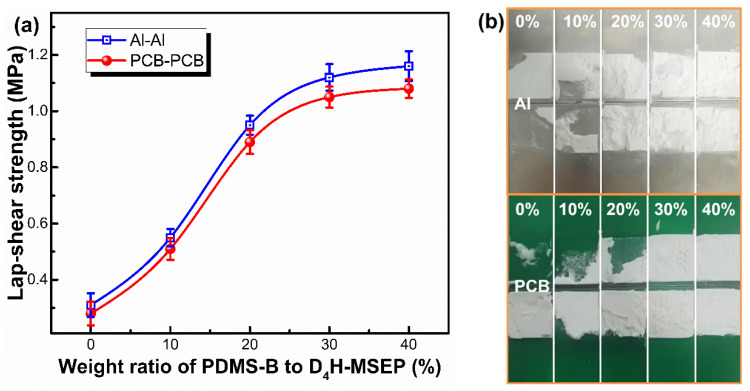
(**a**) Effect of the mass percentage of PDMS-B to D_4_H-MSEP on the lap-shear strength of LSR/Al and LSR/PCB joints, (**b**) optical photographs of LSR/Al and LSR PCB joints after lap-shear strength tests. The mass percentage of PDMS-B to D_4_H-MSEP was 0, 10, 20, 30 and 40%, respectively (LSR with 85.7 wt% Al_2_O_3_ and 2.0 wt% D_4_H-MSEP; curing temperature: 80 °C).

**Figure 7 materials-15-05211-f007:**
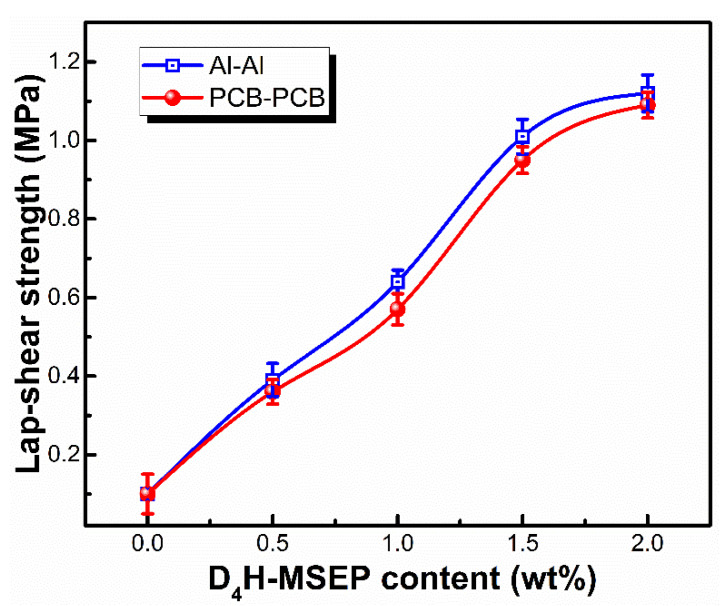
Effect of D_4_H-MSEP content in LSR on the lap-shear strength of LSR/Al and LSR/PCB joints (LSR with 85.7 wt% Al_2_O_3_; curing temperature: 80 °C; PDMS-B/D_4_H-MSEP = 30.0 wt%).

**Figure 8 materials-15-05211-f008:**
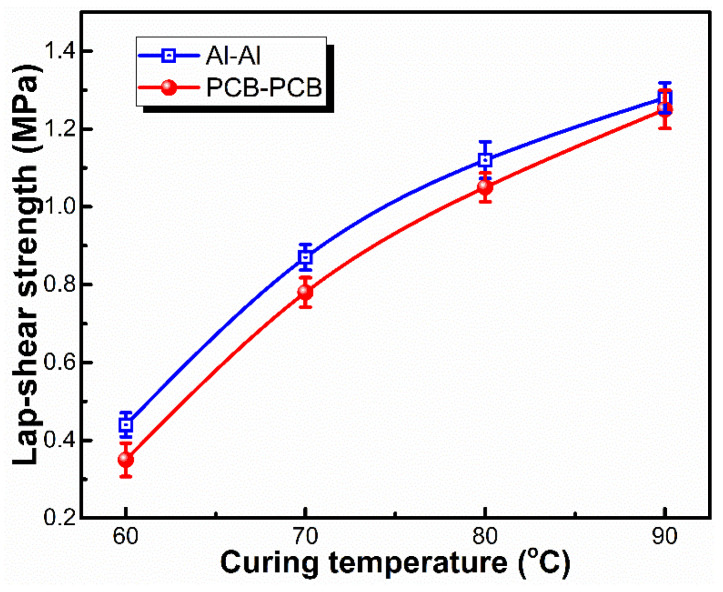
Effect of curing temperature on the lap-shear strength of LSR/Al and LSR/PCB joints (LSR with 85.7 wt% Al_2_O_3_, 0.6 wt% PDMS-B and 2.0 wt% D_4_H-MSEP; curing temperature: 80 °C).

**Figure 9 materials-15-05211-f009:**
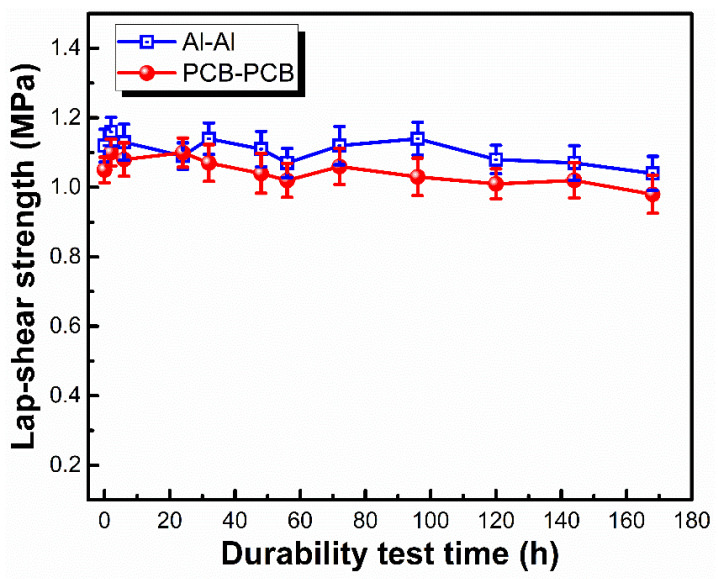
Lap-shear strength of LSR with the durability test time at 85 °C/85% RH condition (LSR with 85.7 wt% Al_2_O_3_, 0.6 wt% PDMS-B and 2.0 wt% D_4_H-MSEP; curing temperature: 80 °C).

**Table 1 materials-15-05211-t001:** Compositions of the two-component LSR.

Compositions	Content (g)	
	Component A	Component B
PDMS-Vi	75.0	66.0
PDMS-mVi	25.0	22.0
PHMS-H	/	12.0
Spherical α-Al_2_O_3_ ^1^	300–700	300–700
PDMS-B	0–9.6	/
D_4_H-MSEP	/	0–32.0
Platinum catalyst	0.8	/
3-methyl-1-pentyn-3-ol	/	1.0

^1^ The mass percentage of Al_2_O_3_ and slicone fluid is 300/100 to 700/100 in both component A and component B, equivalent to the Al_2_O_3_ contents of 75.0 to 87.5 wt% in LSR.

**Table 2 materials-15-05211-t002:** Performance comparison of the LSR prepared in this work with the addition-cured LSR encapsulants reported in the literature and commercial products.

Adhesion Promoter	Substrates	Curing Tempureture	Shear Strength	Ref.
Siloxane oligomers-containing boron and epoxy groups	Cu/Cu	150 °C	3.16 MPa	[30]
Vinyl and epoxy groups-modified silane oligomer	Al/Al	130 °C	1.06 MPa	[32]
Phenyl silicone resin with epoxy and acrylate groups	Al/Al	150 °C	4.43 MPa	[33]
Epoxy, alkoxy and acrylate groups-modified prepolymer	Al/Al	100 °C	0.83 MPa	[34]
KE-1285-A/B ^1^	Al/Al	120 °C	1.50 MPa	[18]
Dow Corning^®^ SE 1816 CV Kit ^2^	Al/Al	150 °C	1.80 MPa	[43]
D_4_H-MSEP/PDMS-B	Al/Al	80 °C	1.12 MPa	This work

^1^ Commercial silicone encapsulant of Shin-Etsu Silicone, Japan. ^2^ Commercial silicone encapsulant of Dow Corning, Midland, MI, USA.

## Data Availability

Not applicable.

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
