# Peer review of "Binary Promoter Improving the Moderate-Temperature Adhesion of Addition-Cured Liquid Silicone Rubber for Thermally Conductive Potting"

_materials, 2022, doi:10.3390/ma15155211_

Round 1

Reviewer 1 Report

Article on the synthesis of adhesion promoters and two-component addition-cured liquid silicone rubber. The paper describes the methodology for synthesizing the materials, the ingredients used, and studies of the parameters characteristic of this group of materials. In my opinion, the authors have made correct introductions and sufficiently described their research, analysis and conclusions. The bibliography seems to be correctly done, and the structure of the article is clear.
The only thing I miss is information on where else, according to the authors, the developed material can be applied because the phenomenon of adhesion is widely used and the demand for such properties presented in this manuscript is desirable.

Author Response

Response: Thanks for your question.

There is a rapidly growing demand for semiconductor-based sensors in application related to optical, electronic devices and electrical modules. These sensors typically contain heat-sensitive substrates and components (such as microelectromechanical systems), which limit the processing temperature of thermally cured encapsulants to a maximum of 80 °C. In addition, encapsulants used in high precise sensors require low stress and warpage to maintain constant and stable performance within the whole operating temperature range. At present, low-temperature cured encapsulants are mainly prepared from epoxy resins, UV-curable polymers and condensation-cured silicone rubbers, while the condensation-cured silicone encapsulants present better electrical insulation, heat and cold resistance, stress relieving properties and long-lasting durability. Moreover, the addition-cured silicone encapsulant shows better thermal conductivity and electrical insulation performance as well as smaller shrinkage compared with those properties of condensation-cured one. Unfortunately, the addition-cured LSR encapsulant shows almost no adhesion to the substrates hence limiting its application. Therefore, the strategy in this work endows LSR with good self-adhesive ability to the substrates at moderate temperature, making it a promising material in the encapsulation of temperature-sensitive electronic devices.

Reviewer 2 Report

In this work, the paper exhibits the Synthesis and use of  Epoxy and alkoxy groups biofunctionalized tetramethylcyclotetrasiloxane (D4H-MSEP)  and boron-modified polydimethylsiloxane (PDMS-B) as synergistic adhesion promoters to provide two-component addition-cured liquid silicone rubber (LSR) with the good self-adhesion ability for applications in electronic packaging at moderate temperature. The synthesized materials were characterized by Fourier transform infrared spectroscopy. The mass percentage of PDMS-B to D4H-MSEP, adhesion promoters content, and curing temperature on adhesion strength of LSR towards substrates were systematically investigated. Additionally, the LSR with 2.0 wt% D4H-MSEP and 0.6 wt% PDMS-B exhibited a lap-shear strength of 1.12  MPa towards Al plates when curing at 80 °C, and the cohesive failure was also observed. The LSR  exhibited a thermal conductivity of 1.59 W m-1 K-1 and good fluidity, which provided sufficient heat dissipation ability and fluidity for potting applications with 85.7 wt% loading of spherical α-Al2O3.  Importantly, 85°C and 85% relative humidity durability testing demonstrated LSR with good encapsulation capacity in long-term processes. Thus, I recommend this manuscript for reconsideration after major revision.

Introduction:

1-      Insert a new paragraph to explain the advantages of the synthesis methodology with respect to other utilizing techniques?

2-      Clarify the benefits of the developed materials that make it the desired choice over others?

3-      Provide some details about the utilized mechanisms and compare them to the other utilized mechanisms?

4-       

Experimental part

Please provide the website addresses of the chemical companies that provide the purchased chemicals?

Results and discussion

1-      The advantages of the developed method for the self-adhesion ability for applications in electronic packaging at moderate temperature in comparison with other methods should be highlighted, including analytical characteristics, reproducibility, specificity, and stability?

2-      With respect to the thermal conductivity of the prepared materials: the excessive incorporation of Al2O3 led to their aggregation, hence the deterioration of LSR mechanical properties The mechanism should be clarified?

3-      Provide more details about the tensile strength and its impact.

4-      The validation of this technique should be introduced by comparison with a previously validated method.

Reviewer 3 Report

This manuscript describes the successful design and synthesis of binary adhesion promoters that provide good adhesion of silicone rubber for protection of electronics components. The synthesis and characterization of the boron-containing and the epoxy and alkoxy functionalized siloxanes are described clearly. In a lap shear test, a sufficient shear strength for practical applications was found with a content of a few percent of a mixture of the adhesion promoters, with a curing temperature of the moderate 80-90 degrees C. The results are of value for practical applications and I recommend the manuscript for publication in Materials without modification.

Author Response

Thank you for acknowledging this research.

Reviewer 4 Report

Interesting results and novelty work. A paper focuses on Binary Promoter Improving the Moderate-Temperature Adhesion of Addition-cured Liquid Silicone Rubber for Thermally Conductive Potting. Though the intention of the authors is highly commendable, there is lot of problems particularly in the presentation throughout the manuscript. Besides there are many grammatical mistakes throughout the manuscript, particularly in respect of use of singular and plural with the subject or verb. In view of the above comments, whole manuscript should be properly written to make it acceptable by Materials journal. I highly recommended this article to be accepted and published in the revised version.

 Abstract:

The abstract given here starts without any background for the present work. Of course, it contains brief details about experimental aspects and the obtained results. However this abstract does not follow the norm of an abstract, which should state briefly:

1.     The purpose of the study undertaken, what are you trying to solve

2.     brief mention of experimental aspects (without using abbreviations)

3.     highlights of the results numerically

4.     Important conclusions based on the obtained results

5.     Potential applications

Therefore, it is suggested that the Abstract to be modified as per the suggestions given above.

 Introduction

Introduction section is long with a many references based on the literature survey conducted by the authors. This is very good. However, this lacks in proper presentation of literature survey, which should have been systematic whereby existing scientific gaps should have been brought out. This should have given justification for the present study, which should be followed by the objectives of this study. In fact there is large amount of literature available on the characterization of Liquid Silicone Rubber. Similarly, a large number of methods to obtain these materials have been used mentioning their advantages and limitations. None of these have been brought out in this study whereby the authors have not justified why they have chosen the method they have used in their study.  It should be noted that normally 'Introduction' should give brief background through literature survey for the study citing previous published work where-by scientific gaps that exist should be brought out. This would have led to justification for the present study.  It is therefore suggested that ‘Introduction Section’ should be revised as suggested above because this Section is an important one from the point of view of taking up the present study.

Relevant article on related matter should be cited such:

Polymers (Basel) 2022;14:1–30. https://doi.org/10.3390/ polym14122432.

Polymers (Basel) 2021;13:2170. https://doi.org/10.3390/polym13132170.

J Mater Res Technol 2021;12:1026–38. https://doi.org/10.1016/j.jmrt.2021.03.020.

Materials (Basel) 2022;15:4062. https://doi.org/10.3390/ma15124062.

Int J Mol Sci 2022;23:6338. https://doi.org/10.3390/ijms23116338.

In my opinion the paper will have good merit if such applications can be demonstrated and reported. Can you give some example?

Please include your objective at the last part of the introduction section.

Materials and Methods:

Normally, this section should have two main subsections. The first one is Materials which should give details of all materials used in the study, where from they were procured, known characteristics, if available (for e.g. Vinyl terminated polydimethylsiloxane etc, where do you get it, what is the purity of the chemical and etc.). Please get all information from the supplier.

The second subsection should be Methods, where methodologies used in the study should be given in a systematic way using sub section with numbers for each of the properties. First the processing or preparation aspects of the final material should be given followed by the characterization of prepared materials including preparation of samples for any specific property or morphology studies should be presented in a systematic way. Here one should also clearly mention the number of samples used, any standards followed for variety of properties, make and model of the instruments used for characterization, their accuracy and experimental conditions used, etc.

It should be known to the authors when one publishes any scientific paper, the results presented therein should be such they should be reproducible by any other person when the experiment is repeated using the same materials. In the present paper, it would be difficult for any other person to repeat the experiments because the chosen materials do not have any pre-history, which is required for other researchers to carryout experiments to check the possible reproducibility of the procedure adopted by these authors.

Some of the paragraph should be under results and discussion and if it is already there then this becomes repetition and hence can be deleted. Secondly, this Section is methods and hence only results should be mentioned and then it should be discussed preferably comparing it with earlier reported similar results by other researchers.

Please separate characterization section 2.5 into several sub section

 Each subsection should cover only one testing and please elaborate it.

Why do you choose content of Al2O3 range 25-88 %? Please justify.

The method is not clear. What is manipulated and constant variables?

Results & Discussion

Well written and easy for the reader to understand what the authors have conveyed.

Some of the paragraph should be under Methods and if it is already there then this becomes repetition and hence can be deleted. Secondly, this Section is Results & Discussion and hence only results should be mentioned and then it should be discussed preferably comparing it with earlier reported similar results by other researchers.

Throughout the manuscript, there are no comparison had been done with other published journal. Therefore, please support your statements with other researcher’s work in the section result and discussion. It should be discussed preferably comparing it with earlier reported similar results by other researchers.

Please insert standard deviation for Durability test.

How many sample did for each experiment? Please do ANNOVA test and standard deviation for all data collected and presented.

Conclusions

Conclusions given here are do not reflect what had been achieved including many speculations. It is too long and should be in 1 paragraph. Hence these need to be suitably modified. It may be remembered that this Section forms a summary of all the major observations/ results obtained. Accordingly, here presentation should consist of the main Results or the observations of the study in short sentences probably with bullet points. This should stand alone or form a subsection of a Discussion or Results Section. Hence better to rewrite this Section based on the comments given in the whole text.

General Comments:

The paper though contains some interesting results and novelty work, it lacks in its proper presentation in the whole manuscript. Of course there is need for better language throughout the manuscript. It is suggested that the authors should take the help of native English speaking person to take care of this problem. In view of these, the paper is highly recommended and should be accepted for publication in the revised form. It is suggested that the authors should revise the paper in the light of above comments/suggestions.

Round 2

Reviewer 2 Report

Authors reply to all comments significantly. Thus,  we recommend the paper to publish in Materials Journal